

# Assessing anxiety, depression and quality of life in patients with peripheral facial palsy: a systematic review

Ferran Cuenca-Martínez[1,2], Eva Zapardiel-Sánchez[1], Enrique Carrasco-González[1], Roy La Touche[1,2,3] and Luis Suso-Martí[2,4]

[1] Departamento de Fisioterapia, Centro Superior de Estudios Universitarios La Salle, Universidad Autónoma de Madrid, Madrid, Spain
[2] Motion in Brains Research Group, Institute of Neuroscience and Sciences of the Movement (INCIMOV), Centro Superior de Estudios Universitarios La Salle, Universidad Autónoma de Madrid, Madrid, Spain
[3] Instituto de Neurociencia y Dolor Craneofacial (INDCRAN), Madrid, Spain
[4] Departament of Physiotherapy, Universidad CEU Cardenal Herrera, CEU Universities, Valencia, Spain

Corresponding author
Roy La Touche,
roylatouche@yahoo.es

## ABSTRACT

**Objective:** Peripheral facial palsy (PFP) is predominantly a unilateral disorder of the facial nerve, which can lead to psychological disorders that can result in decreased quality of life. The aim of this systematic review was to assess anxiety, depression and quality of life symptoms associated with PFP.

**Data sources:** We searched the Medline, PEDro, CINAHL and Google Scholar databases to conduct this systematic review while following Preferred Reporting Items for Systematic Reviews and Meta-Analyses standards. The search was performed by two independent reviewers, and differences between the two reviewers were resolved by consensus.

**Study Selection:** The search terms used were derived from the combination of the following MeSH terms: "facial paralysis", "bell palsy", "anxiety", "anxiety disorders", "depression", "depressive disorders", "quality of life" and not MeSH: "facial palsy", "hemifacial paralysis", "facial paresis", "Peripheral Facial Paralysis", using the combination of different Boolean operators (AND/OR).

**Data Extraction:** On November 1st (2019).

**Data Synthesis:** In total, 18 cross-sectional articles and two case-control studies were selected.

**Conclusions:** The cross-sectional articles showed low methodological quality, while the case-control studies showed acceptable methodological quality. Limited evidence suggests that patients with PFP might have increased levels of anxiety and depressive symptoms. A qualitative analysis also showed limited evidence that quality of life might be diminished in patients with PFP.

**PROSPERO:** CRD42020159843.

## INTRODUCTION

Peripheral facial palsy (PFP) is predominantly a unilateral disorder of the facial nerve (VII cranial nerve), a mixed nerve with both motor and sensory fibers, whose main function is to control the facial expression muscles. In addition, specifically the *chorda tympani* is a uniquely sensitive branch that innervates the anterior two-thirds of the tongue (*Alonso Navarro et al., 2005*; *De Diego-Sastre, Prim-Espada & Fernández-García, 2005*). However, despite being predominantly a unilateral clinical entity, it has been found that PFP can also be a bilateral disorder (*Adour et al., 1978*; *Stahl & Ferit, 1989*).

Peripheral facial palsy has been shown to have several etiologies, which can be classified as (a) idiopathic, commonly referred to as Bell's palsy; (b) infectious, such as Lyme disease, otitis media or even related to herpes virus like the Ramsay Hunt syndrome, which occurs by reactivation of the varicella-zoster virus at the geniculate ganglion; (c) genetic such as Melkersson-Rosenthal syndrome or Albers-Schönberg disease; (d) tumorous; (e) traumatic and (f) post-surgical (*Herrero Velázquez et al., 2009*; *Williams, 2010*; *Lorch & Teach, 2010*).

At the motor level, PFP is related to a loss of essential facial functions, such as blinking, nasal breathing, lip sealing, smiling or speaking (*Roob, Fazekas & Hartung, 1999*; *Lorch & Teach, 2010*). Involvement of the musculature around the eyes is a key clinical finding for differentiating PFP from central facial palsy although at the beginning of the course of an incomplete PFP, the closing of the eyes may not be affecte (*Beurskens & Heymans, 2004*). At the sensory level, PFP is characterized by a diminished sense of taste (*Roob, Fazekas & Hartung, 1999*; *Lorch & Teach, 2010*).

Peripheral facial palsy can lead to psychosocial disorders such as depressive symptoms, high anxiety levels and reduced quality of life. Changes in facial symmetry can lead to depressed mood, which has previously been associated with maladaptive behavior and depressive symptoms (*Macgregor, 1990*; *Valente, 2004*). The facial expressions of patients with PFP are often perceived negatively by observers, even when the patients are smiling (*Ishii et al., 2012*). *Norris et al. (2019)* showed that patients with PFP reported a decline in psychological wellbeing due to their perceived appearance and healthcare experience. Further research is therefore needed to relate the functional disorders of PFP to the possible psychosocial effects that esthetic changes can have on this type of patient. Greater knowledge in this regard could increase the therapeutic efficacy of treatments and the quality of life of patients with PFP.

The main objective of this systematic review was to evaluate the involvement of psychological variables such as anxiety, depressive symptoms and quality of life in patients with PFP.

## MATERIALS AND METHODS

The present systematic review study was conducted with the defined protocol and subdivided into four phases based on the standards of the Preferred Reporting Items for Systematic Reviews and Meta-Analyses statement (Fig. 1) (*Moher et al., 2009*). The protocol of this systematic review was registered in an international register prior to starting the article (PROSPERO, CRD42020159843).

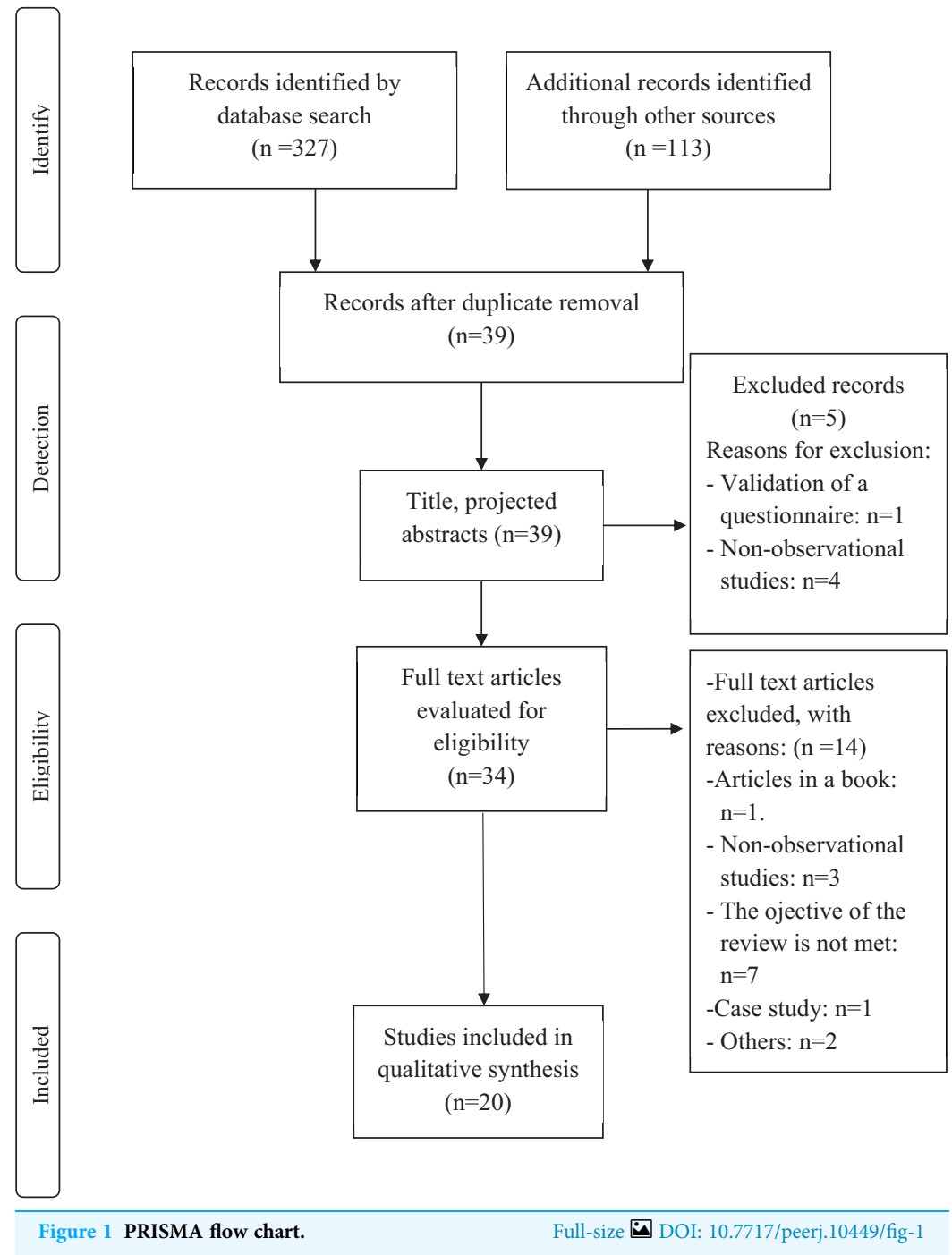

**Figure 1 PRISMA flow chart.**

## Inclusion criteria

The studies included in this systematic review had to meet the following criteria:
(a) cross-sectional, cohort or case-control study with a methodological design;
(b) patients with PFP; and (c) evaluating only the following variables: anxiety, depression and quality of life.

## Search strategy

The systematic search of the included articles was conducted on the following databases: MEDLINE (from 1950 to November 2019), CINAHL (from 1982 to November 2019), PEDro (from 1950 to November 2019) and Google Scholar carried out in the latter on November 1st, 2019. The search terms used were derived from the combination of the following MeSH terms: "facial paralysis", "bell palsy", "anxiety", "anxiety disorders", "depression", "depressive disorders", "quality of life", and not MeSH: "facial palsy", "hemifacial paralysis", "facial paresis", "Peripheral Facial Paralysis", using the combination of different Boolean operators (AND/OR). We used the study design type (observational studies) as a filter.

Two independent reviewers (RLT and FCM) performed the search using the same methodology. Differences between the two reviewers were resolved by consensus.

## Selection criteria and data extraction

Two independent reviewers (EZS and ECG) conducted the first evaluation of the studies to assess the relevance (or lack thereof) of the systematic review. The reviewers used the articles' title, abstract and keywords for this initial screening process. If there was no initial consensus or the information provided by the abstract contained insufficient information, the full text was reviewed.

The full text was then reviewed to ensure that the studies met the inclusion criteria. A third reviewer (LSM) measured when there were differences between the evaluators until a consensus was reached. The data were extracted using a structured protocol that ensured that the most relevant information was obtained from each study.

## Methodological quality assessment

The methodological quality of the articles included in this review was assessed using the Newcastle-Ottawa modified quality assessment scale (NOS) (*Deeks et al., 2003*). NOS is an appropriate tool for assessing the quality of case control, cohort and cross-sectional studies and has moderate interexaminer reliability (*Hootman et al., 2011*). NOS consists of 3 criteria with a minimum of 0 stars and a maximum of 4: participant selection, exposure evaluation, results and comparability. In the total star count, each study is rated as one of the following categories: (1) poor, 0–3 stars; (2) acceptable, 4–5 stars; (3) good, 6–7 stars; and (4) excellent, 8–9 stars.(*Wells et al., 2015*) For the cross-sectional studies, we employed the modified version proposed by *Fingleton et al. (2015)* in which (1) 0–1 out of a possible 3 stars is considered poor quality, (2) 2 out of 3 is considered acceptable quality and (3) 3 out of 3 is considered good quality.

Two independent evaluators (EZS and ECG) reviewed and examined the methodological quality of the studies included in the present study, and disagreements were resolved by consensus through the mediation of a third evaluator (FCM). Interevaluator reliability was determined using the kappa ($\kappa$) coefficient, where (1) $\kappa > 0.7$ indicated a high level of interevaluator agreement, (2) $\kappa = 0.5$–0.7 indicated moderate interevaluator agreement, and (3) $\kappa < 0.5$ indicated low interevaluator agreement (*Cohen, 1960*).

## Qualitative analysis

For the qualitative analysis of the selected studies, we used an adaptation of the classification criteria proposed by *Van Tulder et al. (2003)* for randomized clinical trials adapted for cross-sectional observational, cohort and case-control studies. The results were divided into five levels depending on the methodological quality: (1) strong evidence, representing results from multiple studies (at least three studies); (2) moderate evidence, representing results from multiple low-quality studies and/or one high-quality study; (3) limited evidence, one case-control and/or low-quality cohort study and/or at least two cross-sectional studies; and (4) conflicting evidence, inconsistent findings across multiple studies (at least three studies), and (5) No evidence, where there are no studies of any kind for an outcome category.

## RESULTS

We selected 18 cross-sectional studies (*Weir et al., 1995*; *VanSwearingen et al., 1998*; *VanSwearingen, Cohn & Bajaj-Luthra, 1999*; *Coulson et al., 2004*; *Bradbury, Simons & Sanders, 2006*; *Lee et al., 2007*; *Fu, Bundy & Sadiq, 2011*; *Walker et al., 2012*; *Pouwels et al., 2014*; *Lindsay, Bhama & Hadlock, 2014*; *Sun et al., 2015*; *Kleiss et al., 2015*; *Togni et al., 2016*; *Volk et al., 2017*; *Nellis et al., 2017*; *Worrack et al., 2018*; *Díaz-Aristizabal et al., 2019*; *Muhammad Kassim, Abdullahi & Sammani Usman, 2019*) and 2 case-control studies (*Silva et al., 2011*; *Goines et al., 2016*) in the first phase of the analysis. For all articles, we analyzed the variables depression, anxiety and quality of life. Table 1 presents the study sample's characteristics, the symptom duration, the inclusion criteria, the measures of the variables and their assessment, and the conclusions of the analyzed psychological variables.

## Characteristics of the study population

All studies were conducted with populations with PFP. A total of 2,362 patients were evaluated, with a mean age of 45.7 years (range, 14–81 years). In nine studies, the sample consisted of more women than men (*Weir et al., 1995*; *VanSwearingen, Cohn & Bajaj-Luthra, 1999*; *Bradbury, Simons & Sanders, 2006*; *Walker et al., 2012*; *Pouwels et al., 2014*; *Lindsay, Bhama & Hadlock, 2014*; *Togni et al., 2016*; *Worrack et al., 2018*; *Díaz-Aristizabal et al., 2019*), while two studies had more men than women (*Kleiss et al., 2015*; *Muhammad Kassim, Abdullahi & Sammani Usman, 2019*).

Two studies (*Pouwels et al., 2014*; *Díaz-Aristizabal et al., 2019*) differentiated between patients with left or right PFP. Another study (*Silva et al., 2011*) observed a division between different phases of PFP, including a flaccid phase, a recovery phase and a sequel phase. Finally *Coulson et al. (2004)* and *Muhammad Kassim, Abdullahi & Sammani Usman (2019)* included participants with unilateral lower motor neuron facial nerve paralysis.

## Results of the methodological quality assessment

Of the 20 articles included in this systematic review, 18 were cross-sectional observational studies, and 2 were case-control studies. The 18 cross-sectional studies showed low

**Table 1 Characteristics of the studies included.**

| Article | Design | Characteristics of the sample of PFP group | Characteristics of the sample of control group | Average duration of the symptoms in PFP group | Inclusion criteria in PFP group | Outcome measures | Measurements instruments | General conclusions on psychological variables |
|---|---|---|---|---|---|---|---|---|
| Díaz-Aristizabal et al. (2019) | Cross-sectional study | N = 30; 7 (M); 23 (W) Age: 51,1 (SD = 16,02) Affectation: 12 left; 18 right | – | – | Over 18 years old, minimum evaluation time of 6 month since the injury, incomplete resolution of the PFP and collaboration in the completion of the questionnaires | Degree of PFP deficiency: FGS. Paralysis disability: FDI. Quality of life for patients with PFP | Anxiety and depressive symptoms: HADS | Patients with PFP with higher deficits had greater physical and overall disability and a worse quality of life. However, they did not show greater social disability, nor greater psychological impairment in the form of anxiety and depressive symptoms. Patients with greater physical, social and global disability had greater psychological impairment (anxiety and depressive symptoms) and a worse quality of life. |
| Bradbury, Simons & Sanders (2006) | Cross-sectional study | N = 106 Age: 44,7 34 (M); 72 (W) Form of PFP: 71 aquired paralysis; 35 congenital paralysis | – | At least 12 months after the operation | Patients undergoing vascularized muscle graft reconstruction to correct PFP. The operations had been performed by the same surgeon and the patients were taken from a database at the same hospital | – | Anxiety and depressive symptoms: HADS. Satisfaction degree after PFP: FPEM Other psychosocial variables: semi-structured interview | There is no relationship between dissatisfaction with surgery and anxiety; however, this relationship does occur with depression. Most reported feeling social pressure and distress about their condition. Participants with consistent patterns of avoidance and social isolation, both before and after surgery, were more likely to be depressed than the rest of the study group and more likely to be dissatisfied with the surgery. |
| Coulson et al. (2004) | Cross-sectional study | N = 24 10 (M); 14 (H) Age: 46,1 | N = 24 17 couples, 5 family members, 2 friends | 1 year Mean age: 38,7 | – | Degree of PFP deficiency: Sunnybrook Facial Grading System, Sydney Facial Grading System, HBS. Quality of life: SF-36 PFP disability: FDI. Quality of life after surgery: GBI | Difficulties in facial expression of emotions: questionnaire | A movement deficit associated with the expression of specific emotions and a decrease in quality of life was identified in patients with long-term PFP. |

| Article | Design | Characteristics of the sample of PFP group | Characteristics of the sample of control group | Average duration of the symptoms in PFP group | Inclusion criteria in PFP group | Outcome measures | Measurements instruments | General conclusions on psychological variables |
|---|---|---|---|---|---|---|---|---|
| *Fu, Bundy & Sadiq (2011)* | Cross-sectional study | N = 103 Age: 59 (SD = 17) | – | 6 months—50 years since PFP diagnosis | Patients with a diagnosis of PFP, adequate level of English, PFP longer than 6 months, over 18 years old | Degree of PFP deficiency: HBS Disease beliefs: IPQ-R | Anxiety and depressive symptoms: HADS | Participants have a significant level of psychological distress, which in turn is significantly related to perceptions about their FP. Significant differences were found between the levels of anxiety among women and men. |
| *Muhammad Kassim, Abdullahi & Sammani Usman (2019)* | Cross-sectional study | N = 37 16 (M); 21 (W) Age: 14–70 | – | – | Patients with unilateral PFP over 18 years old with good cognitive ability (more than 24 points in Mini-Mental State Examination) | Degree of PFP deficiency: HBS Community integration of patients with PFP: FAIR | Anxiety and depressive symptoms: HADS | People with PFP may suffer from emotional problems such as the presence of depressive symptoms, social isolation and self-awareness. Women and older people with PFP may be more vulnerable to social isolation. Women may be less depressed than men. |
| *Nellis et al. (2017)* | Cross-sectional study | N = 263 Age: 48,8 | – | PFP developed post-surgery. | Patients without FP that are performed a facial plastic surgery. Older than 18 years old and speak English. Are excluded patients with a previous facial palsy surgery, if they have malignant neoplasm in head or neck, if had head or neck surgery and if couldn't fill the formulary' data on their own | Affectation level of PFPa PFP: HBS Quality of life: EVA (0 = bad, 100 = very good) | Psychometric data: Validated formulary: depression: BDI. Self-reported attractiveness: EVA (0 = I see myself very bad, 100 = I see myself very good). General mood: EVA (0 = very bad; 100 = very good) | Depression significantly greater in patients that developed PFP after surgery. The also saw themselves less good looking, were in a worse mood and presented a decrease in quality of life. There were more women than men depressed. |
| *Kleiss et al. (2015)* | Cross-sectional study | N = 794 40,1 (M); 59,9% (W) Age: 47 (SD = 16) | – | – | Patients with PFP. Children under 14 were excluded | Affectation level of PFP: HBS | Quality of life in patients with PFP: FaCE | Quality of life is more affected in patients with chronic facial paralysis in a psychosocial level. It is also related to increase of the age. |

(Continued)

| Article | Design | Characteristics of the sample of PFP group | Characteristics of the sample of control group | Average duration of the symptoms in PFP group | Inclusion criteria in PFP group | Outcome measures | Measurements instruments | General conclusions on psychological variables |
|---|---|---|---|---|---|---|---|---|
| Volk et al. (2017) | Cross-sectional study | N = 256 103 (M); 153 (W) Age: 52 (SD = 18) | – | – | Patients with unilateral PFP. Patients with bilateral PFP were excluded as well as children under 14 | Affectation level of PFP: HBS. Disability induced by PFP (physical and social): FDI. Quality of life: SF-36 | Quality of life in patients with PFP: FaCE | Patiens with PFP suffer from a severe social and psychological disability. Patients with chronic PFP showed a greater disability than patients with acute PFP. Female an older patients showed more social and psychological disability. |
| Togni et al. (2016) | Cross-sectional study | N = 61 41(W); 20(M) Age: 49.7 (SD = 13,92) | – | – | – | Affectation level of PFP: FGS. Facial disability level: FDI. Temper and character: TCI | Depression: BDI | Personality features modulate facial paralysis disability. Self-management shows up to mediate between the facial paralysis and the disability. Understanding how personality features influence in disability perception can be useful to improve the relationship between the doctor and the patient and to allow personalized and effective therapeutic interventions. |
| Pouwels et al. (2014) | Cross-sectional study | N = 59 Age: 18–75 Affectation: 28 PF left 30 PF right | N = 59 Randomly chosen, without previous PF symptoms, psychiatric or psychological | – | PFP diagnostic, good level of german | – | Anxiety and depressive symptoms: HADS | Significant differences have been found in anxiety and depression between FP patients and healthy controls. Clinically significant differences have not been found in the presence of anxiety or depressive symphoms between right or left PFP. |
| Worrack et al. (2018) | Cross-sectional study | N = 81 | – | – | – | PFP symphtons and sleep evaluation: PSQI Quality of life: SF-36 | Depression: PHQ9. Social Anxiety: LSAS | Facial changes related to PFP causes psychological problems that lead to reduccion in sleep quality. |
| Sun et al. (2015) | Cross-sectional study | N = 21 11(M); 10(W) Age: 23–67 | – | – | All the patients werw diagnosed with acoustic neuroma and the surgery was made by the same doctor in all cases | Level of Affectation of PF: HBS and Burres-Fisch. Quality of life: SF-36. Questionary of post-surgery perceptions | Anxiety: SAS. Autoreported depression: SDS | The PFP caused by the treatment of microsurgery can provoke psychological symphthomps. |

| Article | Design | Characteristics of the sample of PFP group | Characteristics of the sample of control group | Average duration of the symptoms in PFP group | Inclusion criteria in PFP group | Outcome measures | Measurements instruments | General conclusions on psychological variables |
|---|---|---|---|---|---|---|---|---|
| *VanSwearingen et al. (1998)* | Cross-sectional study | N = 48 Age: 49 (18–48) (SD = 16,3) | – | – | Older than 18. Had to be part of the facial center of Pittsburgh University | Affectation level of PFP: FGS. Facial disability level: FDI. Subscale of physical and social wellness | Depresión: BDI | In this study of relations between clinical variables in facial neuromotor disorders, the facial deterioration was related with physical and social disability, with psychological disorders that stress the impact in fisical and psychological disability and are conected with the social disability. |
| *VanSwearingen, Cohn & Bajaj-Luthra (1999)* | Cross-sectional study | N = 29 Age: 50,2 (18–81) (SD: 17,0) | – | – | – | Facial disability level: FDI. Measure of facial movement: MSRA. Highest static response, a quantitive measurement of deterioration,auto-notification rate, especific from the región of social and physical disability, related to facial deterioration | Depression: BDI | The smile and physical disability were the main predictors of depression in patients of neuromuscular facial conditions.Global deterioration of facial movement was not a main predictor. |
| *Lee et al. (2007)* | Cross-sectional study | N = 56 30(M); 26(W) Age: 54,9 (SD: 1,7) | – | – | All the patients spoke english. Nobody with facial pathology previous to surgery | Affectation PFP level: HBS | Quality of life in patients with PFP: FaCE | This impact magnitude in quality of life can be not predicted by the severity of facial paralysis, the age or the gender of the patient, the time passed since the surgery or the size of the tumor. |
| *Lindsay, Bhama & Hadlock (2014)* | Cross-sectional study | N = 148 | – | – | Patients with PFP | Facial evaluation by Software: FACE-gram | Quality of life in patients with PFP: FaCE | The treatment of gracilis muscle produces and increase of the quality of life in patients with PF. |
| *Walker et al. (2012)* | Cross-sectional study | N = 126 (2 women for each man) Age: 50,1 (17–93) | – | – | Patients with PFP older than 16 | – | Anxiety and depression: HADS | The severity of the paralysis is what affects to the psychological wellness of the patient and not the presents of the pathology. |
| *Weir et al. (1995)* | Cross-sectional study | N = 20 12(W); 8(M) Age: 41 (15–78) | – | 6 days from PFP—7 years (average of 65 days). | Patients with idiopathic PFP | Perception of the self-image: FSI. General health state: GHQ. Functional disability: FDQ | Anxiety and depression: part of GHQ | Increase in anxiety an depression levels. |

(Continued)

| Article | Design | Characteristics of the sample of PFP group | Characteristics of the sample of control group | Average duration of the symptoms in PFP group | Inclusion criteria in PFP group | Outcome measures | Measurements instruments | General conclusions on psychological variables |
|---|---|---|---|---|---|---|---|---|
| Silva et al. (2011) | Case-control study | N = 16 5 (M); 11 (W) Age: 43–48 PFP phase: 4 flaccid, 6 recovery y 6 sequel | – | – | Patients with idiopathic PFP older than 18. Patients with traumatic PFP are excluded | PFP phase: otorhinolaryngologic and speech-language exam. Severity level of PFP: HBS | Psychosocial variables: open interview | Patients with PFP in an aftermath phase presented a greater affectation in psychosocial ways followed by those who were in a flaccid and recovery phase. |
| Goines et al. (2016) | Case-control study | N = 16 | N = 4 N = 84 observers 59(W); 25(M) | – | Older than 18, without autism nor schizophrenia | Facial clinic rating scale, quality of life, the paralysis severity and the disability were evaluated by a visual analogical 100 points scale | Effective balance: ABS (happy, content, vigorous, affectionate, anxious, depressed, guilty and hostile) | The observers rated with a less quality of life in patients that presented facial paralysis in comparison to the self-reported disability by the patients. |

**Note:**
PFP, peripheral facial paralysis; M, man; W, woman; SD, standard deviation; FGS, Sunnybrook Facial Grading System; HADS, Hospital Anxiety and Depression Scale; FDI, Facial Disability Index; FaCE, Facial Clinimetric Evaluation; FPEM, Facial Paralysis Evaluation Profile; GBI, Glasgow Befefit Inventory; IPQ-R, Illness Perception Questionnaire-Revised; HBS, House-Brackmann Scale; BDI, Beck Depression Inventory; FAIR, Facial Nerve Palsy Integration Register; VAS, Visual Analogue Scale; TCI, Temperament and Character Inventory; SAS, Self-assessment Anxiety Scale; SDS, Self-assessesed Depression Scale; ABS, Affect Balance Scale; FSI, Facial Self-Image Scale; FDQ, Funcional Disability Questionnaire; GHQ, General Health Questionnaire; MSRA, Measerement of Facial Motion; LSAS, Liebowitz Social Anxiety Scale; PHQ9, Patient Health Questionaire.

methodological quality (with scores of 1 out of 3 stars), while the 2 case-control studies showed acceptable methodological quality (with scores of 6 out of 9 stars). The third evaluator was needed to reach consensus on 2 of the articles. According to the kappa coefficient (κ = 0.736), agreement among the assessors was high. The modified NOS results on the quality of the cross-sectional studies are presented in Table 2, while the results of the quality of the case-control studies are presented in Table 3.

## Qualitative analysis

There was limited evidence that anxiety levels are higher in patients with PFP (*Weir et al., 1995*; *Bradbury, Simons & Sanders, 2006*; *Fu, Bundy & Sadiq, 2011*; *Silva et al., 2011*; *Walker et al., 2012*; *Pouwels et al., 2014*; *Sun et al., 2015*; *Worrack et al., 2018*; *Díaz-Aristizabal et al., 2019*; *Muhammad Kassim, Abdullahi & Sammani Usman, 2019*). This variable was assessed with the Hospital Anxiety and Depression Scale in 6 studies (*Bradbury, Simons & Sanders, 2006*; *Fu, Bundy & Sadiq, 2011*; *Walker et al., 2012*; *Pouwels et al., 2014*; *Díaz-Aristizabal et al., 2019*; *Muhammad Kassim, Abdullahi & Sammani Usman, 2019*), with the Self-reported Anxiety Scale in 1 study (*Sun et al., 2015*), with the General Health Questionnaire in 1 study (*Weir et al., 1995*), by open interview in 1 study (*Silva et al., 2011*) and with the Liebowitz Social Anxiety Scale in 1 study (*Worrack et al., 2018*).

There was limited evidence that depressive symptoms are present in patients with PFP (*Weir et al., 1995*; *VanSwearingen et al., 1998*; *VanSwearingen, Cohn & Bajaj-Luthra, 1999*; *Bradbury, Simons & Sanders, 2006*; *Fu, Bundy & Sadiq, 2011*; *Silva et al., 2011*; *Walker et al., 2012*; *Pouwels et al., 2014*; *Sun et al., 2015*; *Worrack et al., 2018*; *Díaz-Aristizabal et al., 2019*; *Muhammad Kassim, Abdullahi & Sammani Usman, 2019*). One study found that impaired smiling and physical disability might be the main predictors of symptoms compatible with depression. Depressive symptoms was assessed with the Hospital Anxiety and Depression Scale in 6 studies (*Bradbury, Simons & Sanders, 2006*; *Fu, Bundy & Sadiq, 2011*; *Walker et al., 2012*; *Pouwels et al., 2014*; *Díaz-Aristizabal et al., 2019*; *Muhammad Kassim, Abdullahi & Sammani Usman, 2019*), with the Patient Health Questionnaire-9 in 1 study (*Worrack et al., 2018*), with the Self-reported Depression Scale in 1 study (*Sun et al., 2015*), with the Beck Depression Inventory in 2 studies (*VanSwearingen et al., 1998*; *VanSwearingen, Cohn & Bajaj-Luthra, 1999*) with the General Health Questionnaire in 1 study (*Weir et al., 1995*) and by open interview in 1 study (*Silva et al., 2011*).

There was limited evidence that quality of life might be reduced in patients with PFP (*Lee et al., 2007*; *Lindsay, Bhama & Hadlock, 2014*; *Kleiss et al., 2015*; *Goines et al., 2016*; *Nellis et al., 2017*). Quality of life was assessed with the Facial Clinical Evaluation Scale in 2 studies (*Lee et al., 2007*; *Lindsay, Bhama & Hadlock, 2014*) with a 100-point analog scale in 1 study (*Goines et al., 2016*), with a visual analog scale in 1 study (*Nellis et al., 2017*) and with the House-Brackmann scale in 1 study (*Kleiss et al., 2015*).

**Table 2** Quality assessment of cross-sectional studies.

| Cross-sectional studies | S1: Representativeness of the exposed cohort | S2: Selection of the non-exposed cohort | S3: Ascertainment of exposure | S4: Demonstration that outcome of interest was not present at the start of the study | C1a: Study controls for previous injury | C1b: Study controld for age | O1: Assessment of outcome | O2: Follow-up long enoght for outcomes to occur | O3: Adecuacy of follow up of cohorts | Total | % |
|---|---|---|---|---|---|---|---|---|---|---|---|
| *Díaz-Aristizabal et al. (2019)* | – | – | ★ | – | – | – | – | – | – | 1/3 | 33 |
| *Bradbury, Simons & Sanders (2006)* | – | – | ★ | – | – | – | – | – | – | 1/3 | 33 |
| *Coulson et al. (2004)* | – | – | ★ | – | – | – | – | – | – | 1/3 | 33 |
| *Fu, Bundy & Sadiq (2011)* | – | – | ★ | – | – | – | – | – | – | 1/3 | 33 |
| *Muhammad Kassim, Abdullahi & Sammani Usman (2019)* | – | – | ★ | – | – | – | – | – | – | | |
| *Nellis et al. (2017)* | – | – | ★ | – | – | – | – | – | – | 1/3 | 33 |
| *Kleiss et al. (2015)* | – | – | ★ | – | – | – | – | – | – | 1/3 | 33 |
| *Volk et al. (2017)* | – | – | ★ | – | – | – | – | – | – | 1/3 | 33 |
| *Togni et al. (2016)* | – | – | ★ | – | – | – | – | – | – | 1/3 | 33 |
| *Pouwels et al. (2014)* | – | – | ★ | – | – | – | – | – | – | 1/3 | 33 |
| *Worrack et al. (2018)* | – | – | ★ | – | – | – | – | – | – | 1/3 | 33 |
| *Sun et al. (2015)* | – | – | ★ | – | – | – | – | – | – | 1/3 | 33 |
| *VanSwearingen et al. (1998)* | – | – | ★ | – | – | – | – | – | – | 1/3 | 33 |
| *VanSwearingen, Cohn & Bajaj-Luthra (1999)* | – | – | ★ | – | – | – | – | – | – | 1/3 | 33 |
| *Lee et al. (2007)* | – | – | ★ | – | – | – | – | – | – | 1/3 | 33 |
| *Lindsay, Bhama & Hadlock (2014)* | – | – | ★ | – | – | – | – | – | – | 1/3 | 33 |
| *Walker et al. (2012)* | – | – | ★ | – | – | – | – | – | – | 1/3 | 33 |
| *Weir et al. (1995)* | – | – | ★ | – | – | – | – | – | – | 1/3 | 33 |

**Note:**
S, selection; C, comparability; O, outcome; ★, present; no star, not present.

**Table 3 Quality assessment of case studies and controls.**

| Case-control studies | S1: Definition of the cases | S2: Representativiness of the cases | S3: Selection of controls | S4: Definition of controls | C1a: Study controls for previous injury | C1b: Study controls for age | E1: Ascertainmet of exposure | E2: Same method of ascertainment for cases and controls | E3: Non-response rate | Total | % |
|---|---|---|---|---|---|---|---|---|---|---|---|
| *Silva et al. (2011)* | ★ | | ★ | ★ | ★ | ★ | ★ | | | 6/9 | 67 |
| *Goines et al. (2016)* | ★ | | ★ | ★ | ★ | ★ | ★ | | | 6/9 | 67 |

**Note:**
S, selection; C, comparability; E, Exposure; ★, present; no star, not present.

# DISCUSSION

The main objective of this review was to analyze the involvement of psychological variables in PFP. The results showed that anxiety and symptoms compatible with depression can be present in patients with PFP. Quality of life can also be diminished in these patients and is related to the presence of psychological factors.

Psychological factors can play a role in patients with PFP for several reasons, one of which is the possible involvement of facial palsy in an individual's self-concept. The functional and motor disorders of the facial region present in these patients significantly affect their perception of their self-image (*Cross et al., 2000*). This perception of physical attributes and self-image (or self-concept) significantly influences mood and self-esteem (*Campbell, Assanand & Di Paula, 2003*).

The self-concept of patients with PFP is disrupted, in other words, there is a dissonance between their perceived image (the image altered by the PFP) and their learned self-concept, which has previously been associated with a decrease in self-esteem, the presence of anxiety and the appearance of depressive symptoms (*Campbell, Assanand & Di Paula, 2003*). Previous studies have shown that patients with facial palsy have an altered visualization of the affected side and that their self-concept is jeopardized, which could be related to the results of this study (*Ishii et al., 2011*).

Patients with PFP routinely report psychological distress due to the negative implications of PFP in social interactions. Previous studies have indicated that these patients' physical appearance reduces their self-esteem and attractiveness, which, due to social stereotypes, makes them appear less intelligent or confident, leading many people to avoid social relationships with patients with PFP (*Ishii et al., 2012*). Many patients with PFP are unable to express themselves or recognize facial expressions, which can also have a negative effect on their ability to communicate in their social setting, accentuating their difficulties interacting with others (*Korb et al., 2016*).

These patients might therefore view social interactions negatively, leading them to limit or avoid completely such interactions, resulting in social isolation, a factor highly related to poorer moods and depressive symptoms (*Macgregor, 1990*; *Ishii et al., 2016*). A number of studies have shown an association between greater severity of facial palsy and greater depressive symptoms related to these social aspects (*Nellis et al., 2017*).

Some types of facial palsy such as Bell's palsy have been linked to immune system responses in nerve tissue. Previous research studies have shown increased proinflammatory cytokine levels in patients with this disease, which has been linked as a pathogenetic factor to the origin and maintenance of the disease (*Yilmaz et al., 2002*). Elevated anxiety and stress levels can also increase inflammatory levels, which in turn can exacerbate symptoms. For example, one study showed a two-way relationship between anxiety levels and Bell's palsy (*Tseng et al., 2017*).

Lastly, quality of life is related to the subjective perception of health, which includes the relationship between physical and mental health, as well as functionality. Quality of life therefore includes individual satisfaction with the state of health and the emotional response associated with it. Therefore, the aspects mentioned above can have a significant influence on the quality of life of patients with PFP (*Taşkapilioğlu & Karli, 2013*). Our results show that the quality of life can be diminished in these patients and that their perception is closely linked to the psychological factors mentioned above. These results are similar to those obtained for other clinical entities such as temporomandibular disorders (*De Trize et al., 2018*) and chronic migraine (*Fernández-Concepción & Canuet-Delis, 2003*).

From a clinical point of view, the results of the present review emphasize the need to thoroughly assess psychological aspects such as anxiety and depressive symptoms in patients with PFP. The correct management of these factors from a biopsychosocial paradigm can have a major impact on therapeutic success and an increase in the patients' quality of life.

## Limitations

There are several limitations to be considered when interpreting the results of this systematic review. Firstly, the main limitation to be considered is the heterogeneity of the sample included in this article. It is very complicated to draw solid conclusions when you include patients with different states, moments and durations of PFP. For example, both *Bradbury, Simons & Sanders (2006)* and *Coulson et al. (2004)* include patients with PFP lasting at least 1 year. The work carried out by *Díaz-Aristizabal et al. (2019)* includes patients with PFP between 6 and 752 months. *Fu, Bundy & Sadiq (2011)* older than 6 months. *Lee et al. (2007)* deals with patients with PFP secondary to surgery, same as *Sun et al. (2015)*, and with at least 24 months of duration, *Lindsay, Bhama & Hadlock (2014)* includes flaccid PFP or *Silva et al. (2011)* had a sample containing patients with PFP in the flaccid phase, another sample in the recovery phase and another in the sequelae phase. This limitation is very important to consider because the conclusions are established using the whole of this very heterogeneous sample. The results must therefore be considered with great caution. This fact also makes it impossible to carry out a statistical aggregation. Secondly, probably a greater number of psychological variables would have provided more information and could be considered a limitation. Thirdly, a number of the studies presented a high risk of bias due to their methodological quality or limited sample sizes, which could affect the results. Finally, the low number of articles could also be

considered a limitation due to the impossibility of performing a meta-analysis or only because of the difficulty in drawing solid conclusions.

## CONCLUSIONS

The results of this systematic review showed that patients with PFP can have increased anxiety levels and depressive symptoms, as well as having a decrease of quality of life. The quality of evidence of all outcome measures was limited. It is therefore that the presence of these psychological variables in patients with PFP should be considered clinically. However, the authors stress the heterogeneity of the sample and therefore the findings should be taken with caution.

## ACKNOWLEDGEMENTS

The authors would like to thank Centro Superior de Estudios Universitarios CSEU La Salle for its services in editing this manuscript.

### Funding

The authors received no funding for this work.

### Competing Interests

The authors declare that they have no competing interests.

### Author Contributions

- Ferran Cuenca-Martínez conceived and designed the experiments, performed the experiments, analyzed the data, authored or reviewed drafts of the paper, and approved the final draft.
- Eva Zapardiel-Sánchez performed the experiments, prepared figures and/or tables, authored or reviewed drafts of the paper, and approved the final draft.
- Enrique Carrasco-González performed the experiments, prepared figures and/or tables, authored or reviewed drafts of the paper, and approved the final draft.
- Roy La Touche conceived and designed the experiments, performed the experiments, analyzed the data, authored or reviewed drafts of the paper, and approved the final draft.
- Luis Suso-Martí performed the experiments, analyzed the data, prepared figures and/or tables, authored or reviewed drafts of the paper, and approved the final draft.

### Data Availability

There is no raw data; this is a systematic review.

### Supplemental Information

Supplemental information for this article can be found online at http://dx.doi.org/10.7717/peerj.10449#supplemental-information.

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
