# Peer review of "Assessing anxiety, depression and quality of life in patients with peripheral facial palsy: a systematic review"

_PeerJ, doi:10.7717/peerj.10449_

## Round 0.1 · original submission · Major Revisions

Dear Authors,

Please do the major revisions needed for this manuscript.

Thanks

Reviewer 1 ·

Basic reporting

English: needs some minor revision, to involve a native speaker would be helpful.
Literature: In the Introduction, old/outtaken literature is cited, there is newer epidemiological data. Reading the introduction, I get the impression, that the authors themselves do not treat PFP patients. Help from a physician doing so, would help to improve the introduction.

The reviews was focused on depression and anxiety, many psychological variables were not analyzed. This should become clearer in title/abstract/text.

Raw data was not shared, but is not imported for such a narrative systematic review.

Experimental design

The results/conclusions are nor surprising.
A reader knowing the field knows that mainly simple cross-sectional studies on the topic are available and actually no relevant comparing group studies are published.
Hence, the result of such a systematic review is clear before analysis:
Quality of the data is very poor and limited.

And see 1: This narrative review is only about anxiety and depression. Many aspects of the psychological impairment of the patients were not included in the search strategy and analysis.

Validity of the findings

Due to my comments in 2, the impact and novelty of this review is very low.

Actually, NO general conclusions can be made, as the data quality and heterogeneity of the included studies was very low.

Additional comments

This narrative review is about anxiety and depression symptoms in patients with PFP.
The study situation in this field is more than poor. There are NO prospective trial, there are nearly NO comparative group studies. The patients included are so heterogeneous that that this review does not allow to conclude anything.

Abstract: “Peripheral facial palsy (PFP) is a unilateral disorder“ –is not true, PFP can also be bilateral.
Abstract: “Data Extraction: On November 1st.” – the year is missing
Abstract: English style: “The aim of this systematic review was to assess the involvement of various psychological variables associated with PFP.” – a patient can have psychological symptoms but not variables. A study uses variables.
Introduction: “Peripheral facial palsy (PFP) is a unilateral disorder” – see above
Introduction. “and innervate the anterior two-thirds of the tongue” misleading, this is chorda tympani function and only sensory function, no motor fuction.
Introduction: “which can be classified as a) idiopathic, the most prevalent and includes Bell’s palsy and Ramsay Hunt syndrome” – wrong Bell is a synonym for idiopathic, Ramsay-Hunt is NOT an idiopathic form, it is related to herpes virus reactivation. Including a physician dealing with such patients would have been a help.
Introduction: “Involvement of the musculature around the eyes is a key clinical finding for differentiating PFP from central facial palsy”- 1) Imprecise, and in incomplete PFP, orbicularis oculi function can be maintained.
Introduction: “PFP affects 20–32 adults per 56 100,000 per year and 2.7 children per 100,000 per year (Peitersen, 2002; Rowlands et al., 2002) 57 with an annual incidence rate of 11.5–53.3 per 100,000 inhabitants (Martyn & Hughes, 1997).” – if you want present epidemiological data, you should cite epidemiological (!) papers and actual papers. Peitersen 2002, for instance, is cited and cited, but has many methodological flaws and is NOT an epidemiological paper.
Introduction: “control of salivary and tear glands” the patients do not have any loss of control of SALIVARY glands.
Introduction: “as to evaluate the involvement of psychological variables such as anxiety” see above.
Methods: The search concerning psychological impairments of the patients was limited to depression, and anxiety. Therefore, the conclusion is not surprising: “Limited evidence suggests that patients with PFP might have increased levels of anxiety and depressive symptoms” as other psychological factors were not analyzed. What about self-esteem, self-perception, body disfigurement/figurement, psychological/emotional/social stress/distress? – Hence the title, and description of the focus in not correct: “Assessing psychological variables in patients with peripheral facial palsy: A systematic review” – This review was dealing only with anxiety and depression.
Results/Discussion: One important aspect of the heterogeneity of the studies is completely neglected: The duration of the PFP, or better the interval to onset. In some of the studies, patients in the acute phase = flaccid palsy are investigated, in some other studies, many patients have reached the post-paralytic phase and complain mainly of synkinetic/hyperkinetic movements.These are completely different situations with different psychological impact. Furthermore, the underlying etiology may also have a major impact: A cancer patient or a trauma patient can have several other reasons to be anxious or the be depressed.

Reviewer 2 ·

Basic reporting

Lines 60-62: Needs further explanation and specificity regarding eye musculature invovement
Lines 62 - 63: Not completely acurate information. Taste can be altered and post auricular pain sensations can occur

Experimental design

-Medline, PEDro, CINAHL and Google Scholar are a limited band of search engines to perform a systematic review
-The main search term 'facial nerve paralysis' was omitted. Suggest should be updated
-Lines 31-33 Seach terms would have benefitted from including Ramsay Hunt, Herpes Zoster Oticus,
-Line 32 'and not MeSH:'etc unclear sentances

- Although 'bias' is noted in the Limitations section line 248, suggest authors further consider bias as it relates to the articles in the review

Validity of the findings

Lines 250-252: incorrect information presented in the review. Coulson 2004 and Muhammad Kassim N, Abdullahi A, Sammani Usman J. 2019 were not a retrospective studies design and these article stated participants had unilateral lower motor neuron facial nerve paralysis. Please suggest authors thoroughly review and check the accuracy of the ways in which all articles are discussed in their review

The conclusion is somewhat contradictory. The authors report that there is limited evidence and the that patients with PFP can have increased anxiety levels and depressive symptoms and their is limited QoL evidence, and yet they say that results show the clinical relevance of the assessment and correct management of psychological factors.
Please rewite to improve clarity and continuity

---

## Round 0.2 · Major Revisions

Dear Authors,

Please revise the manuscript according to the peer reviewer comments.

Reviewer 1 ·

Basic reporting

The authors addressed many of the queries but not all.

Experimental design

The authors addressed all queries concerning the experimental design but not all answers are convincing:

Query: "Methods: The search concerning psychological impairments of the patients was limited to depression, and anxiety. Therefore, the conclusion is not surprising: “Limited evidence suggests that patients with PFP might have increased levels of anxiety and depressive symptoms” as other psychological factors were not analyzed. What about self-esteem, self-perception, body disfigurement/figurement, psychological/emotional/social stress/distress? – Hence the title, and description of the focus in not correct: “Assessing psychological variables in patients with peripheral facial palsy: A systematic review” – This review was dealing only with anxiety and depression."

Query: "Results/Discussion: One important aspect of the heterogeneity of the studies is completely neglected: The duration of the PFP, or better the interval to onset. In some of the studies, patients in the acute phase = flaccid palsy are investigated, in some other studies, many patients have reached the post-paralytic phase and complain mainly of synkinetic/hyperkinetic movements. These are completely different situations with different psychological impact. Furthermore, the underlying etiology may also have a major impact: A cancer patient or a trauma patient can have several other reasons to be anxious or the be depressed."

It is not enough the say in the Discussion/Limitation that the data is heterogeneous.
The aspects of the two queries should be addressed with more clarity in the Discussion/Limitations because this is of CLINICAL importance and helps the reader to classify the meaning of the results for clinical routine.

Validity of the findings

See. 2.
The limitations have to be addressed more clearly.


My main comment on the findings remains unchanged:
The novelty of this review is low. Actually, NO general conclusions can be made, as the data quality and heterogeneity of the included studies was very low.

Additional comments

Some queries were not answered adequately.

Query: "Introduction: “control of salivary and tear glands” the patients do not have any loss of control of SALIVARY glands.
Response: you are talking about very specific details. That's not what the review is about. Besides, I can easily argue with you because I use the scientific articles to extract the information.
Look at this: “Because of the efferent parasympathetic branches to the lacrimal and salivary glands and the motor branch to the stapedius, additional symptoms such as hyperacusis, decreased lacrimation, or decreased taste can help in localization of the lesion along the course of the facial nerve.”

Decreased lacrimation is related to the affected tear function but SALIVARY function is normal. There is no primary original literature on patients with peripheral facial palsy available showing an impaired salivary function. This is the reason why I recommended to show the manuscript to a physician dealing with such patients.

Query: "Introduction: “Involvement of the musculature around the eyes is a key clinical finding for differentiating PFP from central facial palsy”- 1) Imprecise, and in incomplete PFP, orbicularis oculi function can be maintained.
Response: Our argument is drawn from the state of the art. You offer clinical brushstrokes without providing any reference…look at this paragraph taken from a paper ..."

Here again, it would have been helpful to show the manuscript to a physician dealing with such patients. Many patients with peripheral facial patients present at onset with an incomplete palsy, i.e., for instance, eye closure can be unaffected. You find data on patients with incomplete palsy in any large series or trial on peripheral facial palsy:

Two examples, RCTs, incompleteness is also an important prognostic factor:

Sullivan et al. N Engl J Med. 2007 Oct 18;357(16):1598-607. doi: 10.1056/NEJMoa072006.
Engström et al. Lancet Neurol. 2008 Nov;7(11):993-1000. doi: 10.1016/S1474-4422(08)70221-7. Epub 2008 Oct 10.

---

## Round 0.3 · accepted · Accept

Thank you - your paper has been accepted.

Reviewer 1 ·

Basic reporting

Now, finally, all queries are answered.

Experimental design

Now, finally, all queries are answered.

Validity of the findings

Now, finally, all queries are answered.

Additional comments

Now, finally, all queries are answered.